# SIRT2i_Predictor: A Machine Learning-Based Tool to Facilitate the Discovery of Novel SIRT2 Inhibitors

**DOI:** 10.3390/ph16010127

**Published:** 2023-01-14

**Authors:** Nemanja Djokovic, Minna Rahnasto-Rilla, Nikolaos Lougiakis, Maija Lahtela-Kakkonen, Katarina Nikolic

**Affiliations:** 1Department of Pharmaceutical Chemistry, Faculty of Pharmacy, University of Belgrade, Vojvode Stepe 450, 11221 Belgrade, Serbia; 2School of Pharmacy, University of Eastern Finland, P.O. Box 1627, 70210 Kuopio, Finland; 3Laboratory of Medicinal Chemistry, Section of Pharmaceutical Chemistry, Department of Pharmacy, School of Health Sciences, National and Kapodistrian University of Athens, Panepistimiopolis-Zografou, 15771 Athens, Greece

**Keywords:** virtual screening, Python GUI application, QSAR, machine learning, SIRT2 inhibitors, regression, classification

## Abstract

A growing body of preclinical evidence recognized selective sirtuin 2 (SIRT2) inhibitors as novel therapeutics for treatment of age-related diseases. However, none of the SIRT2 inhibitors have reached clinical trials yet. Transformative potential of machine learning (ML) in early stages of drug discovery has been witnessed by widespread adoption of these techniques in recent years. Despite great potential, there is a lack of robust and large-scale ML models for discovery of novel SIRT2 inhibitors. In order to support virtual screening (VS), lead optimization, or facilitate the selection of SIRT2 inhibitors for experimental evaluation, a machine-learning-based tool titled SIRT2i_Predictor was developed. The tool was built on a panel of high-quality ML regression and classification-based models for prediction of inhibitor potency and SIRT1-3 isoform selectivity. State-of-the-art ML algorithms were used to train the models on a large and diverse dataset containing 1797 compounds. Benchmarking against structure-based VS protocol indicated comparable coverage of chemical space with great gain in speed. The tool was applied to screen the in-house database of compounds, corroborating the utility in the prioritization of compounds for costly in vitro screening campaigns. The easy-to-use web-based interface makes SIRT2i_Predictor a convenient tool for the wider community. The SIRT2i_Predictor’s source code is made available online.

## 1. Introduction

SIRT2 is a NAD^+^-dependent protein deacetylase involved in the regulation of many important biological functions, including the maintenance of genome stability, metabolism, aging, tumorigenesis, and cell-cycle regulation [1,2,3,4,5]. Studies on cellular and animal models of disease revealed the promising potential of SIRT2 inhibition in the treatment of age-related diseases, including neurodegenerative disease and carcinoma [6,7]. Preclinical evidence generated during the last decade has resulted in growing interest in the development of small-molecule SIRT2 inhibitors, particularly as novel anticancer therapeutics [7]. The inhibition of SIRT2 was shown to be an important factor in the treatment of various aspects of tumor development and progression, including inhibition of proliferation, invasion, angiogenesis, and metastatic potential [8,9,10]. In addition to cancer development and progression, SIRT2 was proven to be involved in conferring re-sistance to cancer treatment. Great potential for synergistic combinations of SIRT2 inhibitors with clinically approved drugs was revealed just recently by examining the role of SIRT2 inhibitors in overcoming drug resistance to dasatinib, doxorubicin, or paclitaxel in treatment of melanoma or specific subtypes of breast cancer cells [11,12,13]. Furthermore, the recent study examined selective SIRT2 inhibitors as an augmentation to tumor immunotherapy due to their ability to activate tumor-infiltrating lymphocytes. This approach opened exciting opportunities for future usage of selective SIRT2 inhibitors in overcoming poor clinical response to the TIL (tumor-infiltrating lymphocyte) or CAR-T (chimeric antigen receptor–T cell) immunotherapies [14]. In spite of two decades of vigorous research efforts around the world and many discovered SIRT2 inhibitors, none of the described compounds have entered clinical trials, which signifies the need for novel advances in the field [15]. The most common limitations of known inhibitors includes poor selectivity, potency, or physicochemical properties [10,15,16].

The catalytic core of sirtuins consists of a larger Rossmann-fold domain and a smaller zinc-binding domain connected with several flexible loops (Figure 1A). All sirtuins share the same catalytic mechanism, which involves the formation of a positively charged O-alkylimidate intermediate between NAD^+^ and acetyl–lysine substrate. After several steps, this intermediate hydrolyzes to produce deacetylated polypeptide and 2′-O-acetyl-ADP-ribose (Figure 1B) [17]. Most known inhibitors interfere with this catalytic mechanism by binding to the catalytic site of sirtuins positioned in the cleft between two domains (Figure 1C). Due to the conserved structure of the catalytic site of sirtuins, achieving the selectivity of small-molecule SIRT2 inhibitors turned out to be one of the greatest challenges in the development of this group of compounds (Figure 1C) [15,18]. Recently described pharmacological advantages of selective SIRT2 inhibition over non-selective inhibition of other isoforms of the sirtuin family, particularly sirtuin 1 (SIRT1) and sirtuin 3 (SIRT3), positioned selectivity as one of the most important objectives in development of novel SIRT2 inhibitors [19]. Furthermore, a recent study indicated that the complex conformational behavior of SIRT2 in interaction with inhibitors represents one of the major obstacles in the discovery of novel inhibitors through structure-based computer-aided drug-design (CADD) approaches [20]. However, years of searching for novel SIRT2 inhibitors resulted in large and diverse datasets that could greatly benefit ligand-based CADD approaches relying on specific machine-learning techniques. 

Discovery of novel drugs under the scope of the precision medicine (NIH) initiative heavily relies on integration of large datasets into the drug-discovery pipelines through cheminformatics approaches [21]. The big-data-driven era in modern drug discovery recognized artificial intelligence (AI) as one of the most important tools that could drastically reduce the time and cost of drug discovery in preclinical phases [22,23]. Escalated development and the usage of machine-learning (ML) tools in modern drug discovery was mainly allowed by the availability of large datasets and the democratization of AI. Public databases of pharmacologically active compounds with an ever-increasing number of records on biological activities allowed for a more comprehensive approach to modern drug discovery by the utilization of ML in the modeling of structure–activity relationships [21,22,23]. Quantitative structure–activity relationship (QSAR) modeling is a well-known computational technique for establishing classification-based or regression-based relationships between structural properties of compounds and biological activities [24]. The QSAR technique represents one of the most successful strategies to avoid inactive compounds or to eliminate side effects in pre-clinical drug development [21,22,25]. Retrospective analysis indicated that updating QSAR models as more data become available generally leads to improvements in the accuracy and usefulness of predictions. QSAR models trained on the larger and more diverse datasets are more likely to have a wider applicability domain and to exert a larger coverage of the chemical space. Considering the general improvements in quality as well as broadening the applicability domains of QSAR models trained on larger datasets, global or large-scale QSAR models (i.e., models trained on large data sets of higher compound diversity) are becoming more and more popular [25,26,27]. Currently, there is a lack of large-scale and robust QSAR models for prediction of SIRT2 inhibitor potency and selectivity. Development of such models could aid virtual-screening studies, lead-optimization studies, and repurposing studies, or the integration of cheminformatics with omics data under the more complex precision-medicine pipelines.

With preclinical proof of pharmacological potential of selective SIRT2 inhibitors to treat various modalities of cancer, or to synergize with existing therapies, including immunotherapies, selective SIRT2 inhibitors could prove to be a valuable asset to the existing palette of drugs in the emerging era of personalized medicine. In order to provide open-source computational tools to facilitate the development of SIRT2 inhibitors, in this work we aimed to develop a framework for fast screening and evaluation of novel compounds on SIRT2 inhibitory potency and selectivity. The defined framework, named SIRT2i_Predictor, was built on set of high-quality large-scale classification and regression QSAR models implementing publicly available datasets on selectivity and potency of SIRT2 inhibitors. By creating an appealing and easy-to-use web-based interface, SIRT2i_Predictor was made available to the broader community.

## 2. Results and Discussion

### 2.1. Datasets for Modelling

Available data on the structures and activities of SIRT2 inhibitors were collected from ChEMBL data and literature (see Section 3), resulting in a total of 1797 unique records. Considering the biological-activity measurements (e.g., some of the compounds were tested on additional activities on isoforms SIRT1 or SIRT3, whereas some others were not), the initial pool of data was distributed into four datasets (Datasets 1–4) (Figure 2 and Figure 3 and Table 1). Dataset 1 was intended for building a regression QSAR model, whereas Datasets 2–4 were intended for building different types of classification models. Distributions of activity values (pIC_50_ in Dataset 1) across different activity classes (Datasets 2–4) are depicted in Figure 2, whereas the main characteristics of Datasets 1–4 are summarized in Table 1.

Dataset 1 was built solely on records of compounds with reported SIRT2 inhibitory activity expressed as pIC_50_ values. This dataset contained 1002 compounds with a range of pIC_50_ values from 4 to 7.96. With 1797 entries, Dataset 2 was the largest among the datasets and encompassed all compounds, including activities expressed as both pIC_50_ and Inh%. Considering the criteria presented in Section 3, compounds within Dataset 2 were assigned to the two classes—SIRT2 active and SIRT2 inactive—resulting in almost one-third of compounds assigned as inactive (Figure 2).

Dataset 3 was composed of the compounds that reported inhibitory activities on both SIRT1 and SIRT2 (expressed either as pIC_50_ or Inh%), whereas Dataset 4 was composed of the compounds that reported SIRT2 and SIRT3 inhibitory data, expressed as pIC_50_ or Inh%. Therefore, Datasets 3 and 4 were composed of three classes of compounds: SIRT1(3)/SIRT2 inactive compounds, SIRT2 selective compounds, and SIRT1(3)/SIRT2 nonselective compounds (Figure 2 and Table 1). It should be noted that the bioactivity values used in this study were heterogeneous in origin, determined with different experimental approaches (fluorimetric assay, luminescence assay, electrophoretic mobility shift, scintillation counting) and conditions (time of incubation, acetyl–lysine substrates of different Km values, etc.). In order to make a clearer distinction between classes and eliminate potential noise coming from different experimental conditions, during the creation of Datasets 2–4 compounds with activities lying near fuzzy borders of different classes were omitted. This small group of compounds, referred to in the manuscript as “twilight zone” compounds, had an activity range of IC_50_ = 50–90 µM (for Inh% criteria see Section 3). The recent growth of interest in large-scale QSAR models utilizing datasets from ChEMBL that are heterogeneous in origin has resulted in many studies with similar data-acquisition and -processing strategies as those presented in this work [28,29,30,31,32,33].

Descriptive analysis across different datasets indicated that most of the available compounds obeyed Lipinski’s rule of 5, with several outliers in each dataset (Figure 2) [34]. Compounds from the datasets were pre-processed, and molecular descriptors and fingerprints were calculated using approaches explained in the Section 3. Prior to modeling, all datasets were split into training and test sets using stratified random sampling (70% training set and 30% test set) to ensure the sampling from the same activity distributions. Principal component analysis of different datasets indicated that the splitting strategy was able to maintain equal coverage of the chemical space with training- and test-set compounds (Figure 2). Considering the slight imbalance inside classification datasets (Datasets 2–4), the SMOTE algorithm was used to oversample the minority classes by synthesizing new minority instances prior to training the classification models.

### 2.2. Model Development and Validation

In this study, different regression, binary, and multiclass classification machine-learning models were developed using the combination of five machine-learning algorithms (random forest (RF), support-vector machines (SVM–support-vector classification (SVC) and support-vector regression (SVR)), k-nearest neighbors (KNN), extreme gradient boosting (XGBoost), and deep neural networks (DNN)), as well as four descriptors/fingerprints (Mordred descriptors, ECFP4, ECFP6, and MACCS key fingerprints) (see Section 3). Due to the their easy calculation and the very positive results obtained through cheminformatics studies over the years, recent literature recognized ECFP and MACCS as the most popular ones among commonly used fingerprints [21]. The abovementioned reasons were the discriminatory criteria for the selection of ECFP and MACCS fingerprints in our study, as well. The general workflow of the study is presented in Figure 3. Different regression or classification models for each dataset were trained thorough a process of hyperparameter tuning using Bayesian optimization with five-fold cross-validation. The list of values of hyperparameters used for Bayesian optimization is presented in the Appendix A. For DNN models, more comprehensive optimization of hyperparameters related to the network structure (number of hidden layers, units, and dropouts) was performed using the Keras Tuner with the Bayesian search method. The list of optimized hyperparameters for each combination of modeling approach and descriptor/fingerprint is provided in the Appendix A. The quality of each model trained with optimized hyperparameters was primarily evaluated through an inspection of internal and cross-validation parameters. Afterwards, predictive performance of each model on the “unseen” data was accessed through external validation with the test set. The top-performing models were selected using the consensus approach after evaluating the predictive performance of each model with additional tests. Additional evaluation approaches were chosen depending on the type and purpose of the model. Specificities of each of additional evaluation test are addressed below. The final selected models were considered for constructing the framework of SIRT2i_Predictor (Figure 3).

#### 2.2.1. Regression Models

The combinations of five machine-learning methods with four features (MACCS, ECFP4, and ECFP6 fingerprints, as well as Mordred descriptors [35]) were explored for the development of global regression-based QSAR models using Dataset 1. After the feature-selection procedure (see Section 3), 52 Mordred descriptors were selected for the final QSAR modeling (Appendix A). Fingerprints were used without further reduction of the number of beats. After training the models through the Bayesian hyperparameter optimization procedure with five-fold cross-validation (CV), the quality of each model was accessed initially using internal validation parameters: the coefficient of determination for the training set (*R^2^*) and cross-validated correlation coefficient (*Q^2^*), root mean square error (RMSE) of fitting the training set (*RMSE_int_*), and the RMSE of cross-validation (*RMSE_CV_*). Thresholds for *R^2^* and *Q^2^* were set according to the criteria proposed by Golbraikh and Tropsha (*R^2^* > 0.6, *Q^2^* > 0.5) [36]. Parameters of internal validations, presented in the Appendix A, indicated the internal predictive power, stability, and robustness of each model. Y-scrambling was performed as a part of the internal validation procedure by generating 100 models on randomly shuffled data. The Y-scrambling procedure indicated that the models were not obtained by chance and were highly reliable (Figure 4).

External validation was performed in order to examine the predictive power of the models outside the training data. In alignment with the Organization for Economic Co-operation and Development (OECD) principles, the external validation of the QSAR models could be performed by measuring the goodness-of-fit by coefficient of determination ($R_{ext}^{2}$, should be > 0.6) and root mean square error (*RMSE_ext_* parameter, should be as low as possible) [37,38]. All models performed almost equally well according to these two criteria, with slight predominance in the quality of the models build with ECFP4 and ECFP6 fingerprints (Table 2). However, as discussed by many authors before, relying solely on the simplistic $R_{ext}^{2}$ in some cases could lead to an overoptimistic estimation of the model’s external predictive performance due to the dependence of $R_{ext}^{2}$ on the range of the response values of the test set and their distribution pattern around the training-/test-set mean [39,40,41,42]. Therefore, additional evaluation of the QSAR models’ external predictive power was performed using a set of additional parameters developed accounting for precision (deviation of observations from the fitting line), accuracy (deviation of the regression line from the slope 1 line passing through the origin of Y_observed_ vs. Y_predicted_ curve), and by ensuring that no bias was introduced based on the response scale [40,42]. As additional criteria, the ${\bar{r}}_{m}^{2}$ metrics by Roy et al. (${\bar{r}}_{m}^{2}$ > 0.5 and $\Delta r_{m}^{2}$ < 0.2) [39,43], the $Q_{Fn}^{2}$ metric, and the *CCC* with thresholds proposed by Chirico and Gramatica ($Q_{F1}^{2}$, $Q_{F2}^{2}$, and $Q_{F3}^{2}\;$> 0.7, and *CCC* > 0.85) [40,44] were used (Table 2). Furthermore, the criteria proposed by Golbraikh and Tropsha (($R_{}^{2}$−$R_{0}^{2}$)/$R_{}^{2}$ < 0.1 or ($R_{}^{2}$ − ${R^{\prime }}_{0}^{2}$)/$R_{}^{2}$ < 0.1, 0.85 ≤ k (or k’) ≤ 1.15 and $\left|R^{2}-{R^{\prime }}_{0}^{2}\right|$ < 0.3) were assessed as well (Appendix A) [36]. The created models satisfied almost all of the proposed additional criteria, except $\Delta r_{m}^{2}$ and *CCC*, which failed for some of the models (see Table 2). After discarding the models according to the $\Delta r_{m}^{2}$ and *CCC* criteria, two the most promising models were selected: the XGBoost:ECFP4 model (Figure 4) and the KNN:ECFP6 model (Table 2 and Appendix A).

Following the OECD principles, chemical space boundaries where the model achieved reliable predictions, also known as the applicability domain (AD) of the QSAR model, should be defined as part of external validation and considered when the model is applied for predictions of unknown compounds [37,38,45]. One of the most widely used methods for estimating the boundaries of AD in regression QSAR models is the leverage method [45]. Leverage values are considered proportional to the distances of each compound from the centroid of the training set in the molecular-features space. Leverage values of selected QSAR models are presented through Williams plots (leverage versus standardized residuals) where compounds with leverages and/or residuals above threshold values could be easily detected (Figure 4). Several compounds from the test set were detected to be outside of the applicability domains for the two of the most promising models (XGBoost:ECFP4 and KNN:ECFP6) (Table 2 and Figure 4). The KNN:ECFP6 model was more limited in respect to the coverage of the chemical space, with 39 compounds of the test set being out of the defined AD boundaries compared to the 24 compounds of the XGBoost:ECFP4 model. Additionally, excluding compounds from outside the AD borders resulted in significant improvements in the XGBoost:ECFP4 model statistics compared to those of the KNN:ECFP6 model (Table 2). These results may indicate lower predictive power and lower coverage of chemical space of the KNN:ECFP6 model inside AD borders. Additionally, the KNN:ECFP6 model showed lower robustness through internal validation (Appendix A, cross-validation parameters). Since the global QSAR models are aimed at having broader coverage of chemical space, robustness, and optimal external predictive power, the XGBoost:ECFP4 model was selected for further work. However, it is important to note that regression models were trained on Dataset 1, where most of the compounds were active according to the class assignments (914 active compounds, whereas only 88 compounds were in the “twilight zone” (IC_50_ = 50–90 µM) or “inactive” (IC_50_ > 90 µM)) (see Section 3). This largely limits the usage of regression models only for the pIC_50_ predictions of the active compounds or compounds predicted to be active by classification models, which is further discussed in Section 2.3. Furthermore, the heterogeneity of the data introduced by the numerous sources that populate the ChEMBL database could contribute to the prediction error of the regression models [28]. Therefore, classification-based models that circumvent the abovementioned issue could be expected to have better performance in the identification of active compounds.

#### 2.2.2. Binary Classification Models

According to the general protocol of the study (Figure 3), the combination of five different ML algorithms and four molecular features was explored through Bayes hyperparameter optimization with five-fold CV to develop binary classification models using Dataset 2. The main aim of this part of the study was to train models for the classification of SIRT2 inhibitors and inactive compounds. The rules for the assignment of compounds to the SIRT2 active/inactive class of inhibitors were addressed in the Section 3. Mordred descriptors used for training the binary models were, prior to model training, submitted to the feature-selection protocol (see Section 3 and Appendix A). 233 Mordred descriptors were selected for final modeling. Fingerprints were used without further reduction of the number of beats. 

Parameters of internal validation indicated the internal predictive power, stability, and robustness of each model (Appendix A). The external predictive power of the trained binary models was evaluated using the test set. The following parameters were used for monitoring performances on the external set of different models: balanced accuracy (BA), Matthews correlation coefficient (MCC), area under the receiver operating characteristics curve (ROC_AUC), precision, recall, and F1-score (Table 3). Almost all of the modeling algorithms displayed equally good external predictive power on the external set, with mild predominance of the RF, SVC, and DNN models build using descriptors.

Aiming to recover active molecules from large databases enriched with inactive compounds, virtual screening (VS) is one of the possible real-life applications of this type of ML model. VS models that are able to encompass a larger portion of chemical space are considered more useful since the main objective of screening studies is to find chemically novel and diverse active compounds. Trained on the largest dataset—Dataset 2—the binary model could be expected to have the largest chemical space coverage and to be more useful in VS purposes compared to the models trained on other datasets. In order to additionally evaluate the applicability of selected binary models in the VS, real-life application was simulated by generating almost 20,000 virtual decoy molecules and assigning them to the inactive class. Decoys were created by enforcing 2D topological dissimilarity with known active molecules while retaining similar physical properties. The decoy dataset was merged with an external set, creating the imbalanced database with ratio active:inactive = 1:40. The models were further tested on their ability to recover the active molecules.

In these settings, statistical parameters of the models were recalculated with the addition of early enrichment metrics (Table 4). Early enrichment metrics represent one of the most important parameters for the evaluation of early recognition of VS models since only the top-ranked compounds are usually considered for experimental evaluation. Early recognition is the direct reflection of the models’ ability to rank active molecules very early in an ordered list. Herein, we used ROC EF 0.5%, 1%, 2%, and 5%, which quantified the area covered by the curve at 0.5%, 1%, 2%, and 5% of the screened false positives, respectively [20,46]. With the dataset containing a significantly larger number of chemically diverse inactive compounds, the RF:ECFP4 binary model stood out as the model with the greatest predictive power (Table 4 and Figure 5). In the heavily disbalanced decoy set, the RF:ECFP4 binary model displayed better sensitivity, specificity, precision, and robustness but also better early recognition. In 0.5% of false positives, the RF:ECFP4 binary model was able to find over 70% of true active molecules (Table 4). It is worth noting that the most of the true inactive compounds from Dataset 2 were chemically similar to the active compounds, whereas the decoy dataset was enriched in topologically dissimilar compounds. Since the parameters calculated on the decoy dataset represented the more reliable estimation of the model’s predictive performance in VS settings, the RF:ECFP4 model was selected for further work.

The applicability domain of the selected models was defined according to the indeterminate-zone approach [47,48,49]. Predictions that predicted probabilities falling into indeterminate zones (in-zone predictions) are considered unconfident and vice versa. For binary models, this zone was set to 0.5 ± 0.1 of the prediction probability for corresponding classes. Considering the applicability-domain corrections, the performance of the selected RF:ECFP4 model significantly improved (Table 4 and Figure 5).

#### 2.2.3. Multiclass Classification Models

Being assigned to the same class of sirtuins—Class I—SIRT1 and 3 are the closest homologues to SIRT2, which explains the difficulties in achieving selectivity of SIRT2 inhibitors [50]. With recent links established between the safety of SIRT2 inhibitors and selectivity profiles, achieving the selectivity of novel compounds appears to be a rising trend in the drug-discovery efforts of SIRT2 inhibitors [19]. A notable amount of available structure–activity data on ChEMBL contains molecules with parallel inputs on activity towards different sirtuin isoforms, the most abundant being SIRT1 and 3. However, the size of the subsets of SIRT2 inhibitors with parallel records on SIRT1 or SIRT3 inhibitory activity were still significantly smaller and more imbalanced than the global SIRT2 dataset (Figure 2), which could hamper the predictive power and applicability of models created using these datasets. The main goal of this part of the study was to create and validate models for prediction of the selectivity of the potential inhibitors.

In alignment with the general protocol (Figure 3), all previously mentioned ML algorithms and molecule features were used to build and validate selectivity models. Two types of models were built: a SIRT1/2 selectivity model and a SIRT2/3 selectivity model. Due to the limited number of compounds with concomitant SIRT1/2/3 records accompanied by significant class imbalances, a joint SIRT1/2/3 model was not built. A total of 270 Mordred descriptors was selected for SIRT1/2 selectivity models, whereas 316 Mordred descriptors were selected for SIRT2/3 modeling (Appendix A). Fingerprints were used without further reduction of the number of beats. Selectivity models were built as multi-class prediction models, where different classes represented selective SIRT2, non-selective SIRT1/2 or SIRT2/3 and SIRT1/2, or SIRT2/3 inactive compounds (Figure 2). Similar to the VS binary models, external and internal predictive performance of selectivity models was evaluated using the same set of statistical parameters (Figure 6 and the Appendix A). Internal validation parameters indicated good internal predictive performance of the trained model (Appendix A). Interestingly, for the selectivity models, models built with descriptors performed significantly better on external validation with the test set compared to the models built with topological fingerprints (Appendix A), with a predominance of RF, DNN, and SVC ML approaches. It is more likely that physicochemical properties encoded in selected descriptors for each modeling approach played greater importance in selectivity than structural features of the molecules. It is interesting to note that DNN models achieved the greatest performance across all types of molecular features for the SIRT2/3 selectivity model, which indicates the ability of the deep learning approach to more efficiently learn from the limited number of training data (Dataset 4–the smallest dataset) (Figure 6A).

Practical application of the selectivity models could involve making predictions across the large number of inactive compounds. Although models have been trained to recognize inactive compounds, limited chemical space coverage of true inactive compounds of Datasets 3 and 4 may limit the applicability of these models. In order to emulate behavior in real-life application of these models, when facing the large number of inactive compounds, models have been additionally evaluated on decoy datasets (active: inactive = 1:40) similar to binary models. When enriching the inactive class in a decoy dataset, statistical parameters showed a slight shift in the model’s predictive power. Surprisingly, the decoy-set analysis revealed the advantage of ECFP4 molecule representations in the case of the SIRT1/2 models (Figure 6B and Appendix A). After evaluation of the decoy dataset, the RF:ECFP4 SIRT1/2 model stood out as significantly better compared to the other models. A similar situation was seen in the SIRT2/3 models, where the RF:ECFP4 SIRT2/3 model displayed increased predictive performance on the decoy set. However, in the case of the SIRT2/3 models, the situation was less clear and the DNN:descriptors SIRT2/3 model showed comparable performance (Figure 6B and Appendix A). It should be noted that the SIRT2/3 models generally performed poorly on the highly imbalanced decoy datasets created by maximization of 2D topological dissimilarity of decoy compounds. This limits usage of SIRT2/3 models only on compounds topologically similar to the already-known active compounds. Generally poor performance of SIRT2/3 models on the decoy dataset could be attributed to the limited size and diversity of the dataset used for training.

In order to further explore the applicability of the selected models, applicability domains were defined. Similar to the binary models, the indeterminate-zone approach was used to define AD for the selectivity models. Since our models have three classes as outcomes, the AD zone where predictions are considered confident was defined as >0.5 for the probability of the predicted class. Considering only the data points within AD, the greatest improvement in predictive statistics was observed in the case of the SIRT1/2 model (Figure 6B and Appendix A). On the other hand, two promising SIRT2/3 models with similar performances on the decoy dataset (RF:ECFP4 SIRT2/3 and DNN:descriptors SIRT2/3) displayed just slight improvements after AD corrections (Figure 6B and Appendix A). However, the DNN model had much better coverage, including almost 19,000 compounds (compared to 9000 for RF) within the AD borders, so it was retained for further work.

In summary, the SIRT1/2 model showed excellent predictive power, whereas the SIRT2/3 models demonstrated lower quality on topologically dissimilar compounds. The most probable explanation for this result may be in the size differences of the utilized datasets. Considering the abovementioned limitations in the dataset’s size and diversity as well as the quality of the models, selectivity models SIRT1/2 and SIRT2/3 are the most appropriate to be applied as tools for additional selectivity analysis of the virtual screening results where the activity of the compounds is already predicted with the more accurate binary models. Conflicting predictions (e.g., when binary models judge a compound as active, whereas selectivity models judge a compound as inactive) inside the applicability domain of the models should be addressed with caution, and in that context selectivity models (specifically, the SIRT1/2 model) could be used as additional confirmation of the compound activity.

### 2.3. SIRT2i_Predictor’s Framework for Discovery of Novel Inhibitors

In order to increase the availability and ensure the best practice in the application of the created models, a framework for prediction of the activity/selectivity of novel compounds was defined and encoded into the Python-based application named SIRT2i_Predictor. The workflow of SIRT2i_Predictor is presented in Figure 7 and roughly consists of (1) a module selector, (2) a SMILES preprocessor, (3) predictors, and (4) analyzers. With the aim of making SIRT2i_Predictor easily accessible to the wider community, the appealing web-based graphical user interface (GUI) was created as well (Figure 8).

The module selector allows the user to choose between the two modules of the framework: VS and SMILES-Analyzer modules. In the VS module, SIRT2i_Predictor uses CSV files as an input, which contain compounds in SMILES format with or without compound IDs (up to 200 MB in size). SMILES are prepared for predictions automatically by a SMILES pre-processor. The predictor of the VS module relies solely on the binary-classification RF:ECFP4 model. Compared to the binary model, selectivity models, especially the SIRT2/3 model, demonstrated limited utility on the decoy set, which could be attributed to the limited size and the diversity of the training sets (see Section 2.2.3). On the other hand, regression models, trained mostly on the active compounds, may not be the best choice for VS purposes. Due to the limited presence of inactive compounds in the training set (as discussed above in Section 2.2.1), the regression model could represent a valuable analysis tool for detailed analysis of compounds predicted to be active by the binary model. Therefore, the binary RF:ECFP4 classification model, due to having the largest and the most diverse dataset, as well as superior performance in the real-life application, was selected as the primary virtual-screening model. As an output, the VS module of SIRT2i_Predictors creates and writes the table of screened molecules (CSV output file), together with the predicted probabilities and class assignments (“Yes” class if the molecule is predicted to be an inhibitor, “No” class if not) (Figure 8). This utility could represent a great asset in the prioritization of compounds and cost reductions in large-scale in vitro screening campaigns. 

The other module, SMILES-Analyzer, is intended for more detailed analysis of the VS results. However, it can be used independently of the VS module for analysis of desirable compounds. The SMILES-Analyzer module requires list of SMILES as an input, which are pasted by the user into the input box (Figure 8). The predictor of this module encompass all four ML models (binary RF:ECFP4 model, regression XGBoost:ECFP4 model, selectivity RF:ECFP4 SIRT1/2, and DNN:descriptors SIRT2/3 models), whereas different analyzers provide the user with detailed reports on the predicted potencies and selectivity of predicted inhibitors (Figure 7). The main objective of the SMILES-Analyzer module is to assist with the analysis and selection of VS results for further experimental evaluation.

Similar to the VS module, the SMILES-Analyzer module could also process a large number of compounds. However, due to the different requirements of each model on the inputs, the preprocessing step of this module requires more computational time, which could be problematic for screening very large databases. In addition to the text and numerical summaries of predictions for compounds made by all four ML models, the SMILES-Analyzer module allows for the detailed inspection of each compound independently through the graphical interpretations of the predictions (Figure 8). Graphical interpretation of the predictions includes radar-chart summary of predictions made by all four models, a histogram of predicted probabilities for estimation of the confidences (applicability domain) of classification-model predictions, a leverage plot for estimation of the confidence (applicability domain) of pIC_50_ predictions, and the maps of the per-atom contributions (positive and negative) for predicted SIRT2 activity (with regression and binary models) (Figure 8). Atomic-contribution maps are built according to the concept of similarity maps, where the contribution of each atom is calculated with respect to the differences in predicted probabilities obtained when the bits in the fingerprint corresponding to the atom are removed [51]. Additionally, GUI provides the predictions of the most similar compound in the ChEMBL database using Tanimoto similarity and ECFP4 fingerprints. Tanimoto analysis is provided in order to judge the confidence of SIRT2/3 selectivity predictions since the DNN:descriptors SIRT2/3 model displayed limited applicability on the diverse decoy set (see Section 2.2.3). The SMILES-Analyzer module, similar to the VS module, also outputs a tabular report (CSV file) for predictions on all input molecules.

### 2.4. Benchmarking SIRT2i_Predictor against the Structure-Based VS Approach

The novel structure-based virtual-screening (SBVS) approach relying on alternative conformational states of SIRT2 discovered through computationally intensive simulations of the binding-pocket dynamics was recently published by our group [20]. The utilization of alternative binding-pocket conformational states, besides showing significant improvements in validation metrics compared to the single-structure approach, resulted in the expansion of the chemical space coverage of virtual hits. In order to test SIRT2i_Predictor’s ability to expand the chemical space of virtual hits, we repeated the prospective SBVS campaign from the aforementioned paper encompassing around 200,000 compounds from the SPECS database [52]. Self-organizing maps representing chemical space coverage were constructed and compared between different approaches. SIRT2i_Predictor binary models showed a surprising expansion of chemical space of virtual hits compared to the known chemical space of SIRT2 inhibitors (Figure 9A). Compared to the SBVS approach, the expansion of chemical space was just slightly lower, which places SIRT2i_Predictor as a comparable tool in search of novel chemical scaffolds of inhibitors. To further explore the reach of SIRT2i_Predictor in unexplored portions of chemical space, one of the top-ranked compounds occupying a portion of chemical space not covered either by SBVS or by ChEMBL compounds was singled out and analyzed using the SMILES-Analyzer module. This portion of chemical space was occupied by a thiohydantoin derivative, which was predicted to be a non-selective SIRT2 inhibitor with all probabilities being inside applicability domains (Figure 9B). Structure–activity analysis of SIRT2i_Predictor based on similarity maps revealed the thiohydantoin scaffold as crucial for predicted activity (Figure 9C). To our knowledge, none of the thiohydantoin derivatives had been reported as sirtuin inhibitors, which further corroborated SIRT2i_Predictor’s capacity to provide novel scaffolds of inhibitors and demonstrated its utility in creating structure–activity hypotheses. As expected, the VS module of SIRT2i_Predictor was able to screen 200,000 compounds in a matter of minutes, which was a significant improvement compared to the SBVS campaign, requiring hours.

In our SBVS campaign, nine hit molecules extracted from previously unexplored portions of chemical space were shown to have potency on SIRT2. However, the IC_50_ values of two lead compounds as well as the Inh% of five compounds were in the “twilight zone” of the binary models (IC_50_ = 50–90 µM; Inh%@200 µM = 40–80%), whereas only two compounds were true inactive compounds. In order to explore the performance of SIRT2i_Predictor on this group of “twilight zone” compounds from distant portions of the chemical space of known inhibitors, nine hit compounds were analyzed using the VS module of the framework. The results of the predictions were in agreement with the experimental observations (Appendix A), where SIRT2i_Predictor predicted that none of the compounds would have IC_50_ < 50 μM, which was the criterion for classifying a compound as active in binary models. Most of the compounds were assigned as being outside the applicability domain (5 of 9), which corresponds to the “twilight zone” and the fact that the compounds were selected from distant portions of chemical space in the SBVS study. The rest of the compounds were classified as inactive (4 of 9).

Further analysis of the utility of SIRT2i_Predictor was performed by analyzing our small in-house database of compounds (Appendix A) [53,54]. Unfortunately, predicted probabilities for binary models for all in-house compounds were outside the defined applicability domain or predicted to be inactive. In order to further justify SIRT2i_Predictor as a valuable tool in processing the VS data, we utilized our previously published SBVS model, and four compounds predicted to be active according to the SBVS model were tested in vitro. Although the compounds showed some level of SIRT2 inhibitory activity (Appendix A), all compounds were shown to be poor inhibitors (three were in the “twilight zone” and one was inactive according to the criteria for classification models). These results were in agreement with SIRT2i_Predictor’s estimations, which predicted that the compounds would be either inactive or outside the applicability domain. As a standard inhibitor, we used compound EX-527 (IC_50_ (SIRT2) = 20 µM) [55]. This compound was not included in the training sets for the binary model of SIRT2i_Predictor. SIRT2i_Predictor clearly distinguished EX-527 from the in-house compounds, whereas the predictions for in-house compounds were in agreement with the experimental results.

In summary, SIRT2i_Predictor demonstrated utility in discarding the compounds of lower potency while retaining comparable coverage of chemical space as some of more computationally demanding SBVS approaches. The presented benchmarking results place the SIRT2i_Predictor as a complementary tool to the SBVS approaches, which can be utilized as standalone virtual-screening tool, but also as an additional fast and convenient filter after virtual or in vitro screening studies to further prioritize the most promising compounds for biological evaluations.

## 3. Materials and Methods

### 3.1. Dataset Preparation

The initial database of structures and biological activities was prepared by collecting all records from the ChEMBL database (release 30) with the reported SIRT2 inhibitory activity (expressed as half-maximal inhibitory concentration (IC_50_) values or percent of inhibition (Inh%)) [56]. Additional compounds were acquired from patent US20160376238A1 [57]. The compounds were extracted from the patent document using ChemDataExtractor (v 1.3.0) software [58]. The raw data were initially divided into the four datasets (Datasets 1–4) according to the intended purpose (see below). After collecting the raw data, Datasets 1–4 were manually curated by removing records with activities reported for anything but inhibition of deacetylation reaction (e.g., different defatty-acylation activities, etc.). Curated datasets were further pre-processed by canonizing the SMILES, removing the duplicates, stripping the salts, and unchanging the molecules. All pre-processing steps were performed using the RDKit (v 2021.03.4) [59]. Before removal, duplicated records with multiple activities were manually inspected, and only records with activity closest to the average within the group were retained. If the same molecule contained IC_50_ and %Inh records, the record with IC_50_ values was retained.

For the regression models, the IC_50_ values were converted into pIC_50_ (pIC_50_ = −log10 (IC_50_)), whereas for the classification models the IC_50_ values and Inh% were encoded into different classes. In this study, compounds were assigned to the “SIRT2 active” class if IC_50_ ≤ 50 µM, or Inh% ≥ 80% (if assayed at 200 µM), or Inh% ≥ 70% (if assayed at 100 µM), or Inh% ≥ 60% (if assayed between 50 and 100 µM), or Inh% ≥ 50% (if assayed bellow 50 µM). Compounds were assigned to the SIRT2 “inactive” class if IC_50_ ≥ 90 µM or Inh% ≤ 40% (if assayed above 100 µM). For multiclass models, the same criteria were applied with the records on SIRT1 and SIRT3 activities. In subsequent modeling steps, Datasets 1–4 were further divided into the training (internal) set (70%) and test (external) set (30%) using the stratified train–test split algorithm of the scikit-learn library (v 1.1.1) [60]. Class imbalance before training the classification models was accessed using the SMOTE (Synthetic Minority Oversampling Technique) algorithm from the imbalanced-learn library (v 0.9.1) [61].

### 3.2. Calculation of Molecular Features and Feature Selection

After preparation of the dataset, all molecules were encoded using 166 bit-long MACCS key fingerprints, 1024 bit-long extended-connectivity fingerprints (ECFP4 and ECFP6) as implemented in RDKit, or 1613 two-dimensional descriptors calculated by Mordred (v 1.2.0) [35]. The number of descriptors was further reduced for each dataset independently. Firstly, descriptors with zero or “NaN” values were removed. After standardization, low-variance descriptors were removed using the cut-off value 0.1. Correlations across all pairs of descriptors was calculated using Pearson correlation coefficient. Any two descriptors with correlation values higher than 0.9 were regarded as redundant and only one was retained. The final selection of descriptors for the modeling was performed according to recursive feature elimination with cross-validation (CV), as implemented in Python library scikit-learn (v 1.1.1) [60]. A decision tree classifier was used as an estimator, with 10-fold cross-validation. All descriptors selection steps were performed only for training-set compounds.

### 3.3. Model Building and Evaluation

In this study, five ML algorithms were utilized for model building: random forest (RF), support-vector machines (SVM: support-vector classification (SVC) and support-vector regression (SVR)), k-nearest neighbors (KNN), extreme gradient boosting (XGBoost), and deep neural networks (DNN). Repressors and classifiers for RF, SVM, XGBoost, and KNN models were built using the scikit-learn library with the addition of the XGBoost Python library (v 1.5.1). DNN models were generated using TensorFlow (v 2.9.1) [62]. For RF, SVC (SVR), KNN, and XGBoost hyperparameter tuning was performed with the Bayesian optimizations using the five-fold CV as implemented in the scikit-optimized library (v 0.8.1). For DNN models, hyperparameter optimizations, together with comprehensive optimization of the neural network architectures, was performed with custom Keras Tuner (v 1.1.1) [63] scripts and using the Bayesian optimization with five-fold CV. A detailed list of all hyperparameters optimized, together with objectives and other specificities of ML approaches, are presented in Appendix A.

Briefly, three types of ML models were trained on training sets of corresponding datasets: regression-based (Dataset 1), binary classification-based (Dataset 2), and multiclass classification-based models (Datasets 3 and 4). Internal validation of the regression models was performed using the coefficient of determination for the training set (*R^2^*) and the cross-validated correlation coefficient (*Q^2^*), the root mean square error (RMSE) of fitting the training set (*RMSE_int_*), and the RMSE of cross-validation (*RMSE_CV_*) [64]. Y-scrambling was performed as a part of an internal validation procedure to check the robustness and reliability of top-performing models. Y-scrambling was performed by generating 100 models using randomly shuffled data with the same hyperparameters used for training the initial model. Evaluation of external predictive power of regression-based QSAR models was performed using the set of different validation metrics: determination coefficient of the test set ($R_{ext}^{2}$) (Equation (1)), RMSE of the external set ($RMSE_{ext}$) (Equation (2)) [64], $Q_{Fn}^{2}$ metrics ($Q_{F1}^{2}$, $Q_{F2}^{2}$, $Q_{F3}^{2}$) (Equations (3)–(5)) [37,65,66], $r_{m}^{2}$ metrics ($r_{m}^{2}$, ${\bar{r}}_{m}^{2}$, $\Delta r_{m}^{2}$) (Equations (6) and (7)) [39,43], and the concordance correlation coefficient (*CCC*) (Equation (8)) [40]. Formulas for the calculation of internal validation metrics are presented in the Appendix A.
(1)$$R_{ext}^{2}=1-\frac{\sum_{i=1}^{n_{EXT}}{\left(y_{i}-{\hat{y}}_{i}\right)}^{2}}{\sum_{i=1}^{n_{EXT}}{\left(y_{i}-\bar{y}\right)}^{2}}$$
(2)$$RMSE_{ext}=\sqrt{\frac{\sum_{n=1}^{n_{EXT}}{\left(y_{i}-{\hat{y}}_{i}\right)}^{2}}{n_{EXT}}}$$
(3)$$Q_{F1}^{2}=1-\frac{\sum_{i=1}^{n_{EXT}}{\left(y_{i}-\hat{y_{i}}\right)}^{2}}{\sum_{i=1}^{n_{EXT}}{\left(y_{i}-{\bar{y}}_{TR}\right)}^{2}}$$
(4)$$Q_{F2}^{2}=1-\frac{\sum_{i=1}^{n_{EXT}}{\left(y_{i}-\hat{y_{i}}\right)}^{2}}{\sum_{i=1}^{n_{EXT}}{\left(y_{i}-{\bar{y}}_{EXT}\right)}^{2}}$$
(5)$$Q_{F3}^{2}=1-\frac{\left[\sum_{i=1}^{n_{EXT}}{\left(y_{i}-\hat{y_{i}}\right)}^{2}\right]/n_{EXT}}{\left[\sum_{i=1}^{n_{TR}}{\left(y_{i}-{\bar{y}}_{TR}\right)}^{2}\right]/n_{TR}}$$
(6)$$r_{m}^{2}=r^{2}\left(1-\sqrt{\left(r^{2}-r_{0}^{2}\right)}\right)$$
(7)$$\Delta r_{m}^{2}=\left|r_{m}^{2}-{r^{\prime }}_{m}^{2}\right|$$
(8)$$CCC=\frac{2\sum_{i=1}^{n_{EXT}}\left(y_{i}-\bar{y}\right)\left({\hat{y}}_{i}-\bar{\hat{y}}\right)}{\sum_{i=1}^{n_{EXT}}{\left(y_{i}-\bar{y}\right)}^{2}+\sum_{i=1}^{n_{EXT}}{\left({\hat{y}}_{i}-\bar{\hat{y}}\right)}^{2}+n_{EXT}{\left({\bar{y}}_{i}-\bar{\hat{y}}\right)}^{2}}$$

In Equations (1)–(8), *TR* represents the training set, *EXT* represents the test set (external set), $y_{i}$ represents the experimental data values, ${\hat{y}}_{i}$ represents the predicted data values, ${\bar{y}}_{i}$ represents the average of the experimental data values, and $\bar{\hat{y}}$ represents the average of the predicted data values. $r_{0}^{2}\;$ and $r^{2}$ are the determination coefficients of the regression function calculated using the experimental and predicted data of the external set, forcing the regression through the origin ($r_{0}^{2}$) or not ($r^{2}$). $r_{m}^{2}$ was calculated using the experimental values on the ordinate axis, whereas ${r^{\prime }}_{m}^{2}$ was calculated using the same values on the abscissa. ${\bar{r}}_{m}^{2}$ is the average of $r_{m}^{2}$ and ${r^{\prime }}_{m}^{2}$.
(9)$$h_{i}=x_{i}^{T}{\left(X^{T}X\right)}^{-1}x_{i}$$
(10)$$h^{*}=\frac{3\left(m+1\right)}{p}$$

The applicability domain of the created regression-based models was performed according to the leverage method [45]. The leverage values (*h_i_*) were computed according to Equation (9), where *X* is the matrix formed with rows corresponding to the most important descriptors/bits of molecules from the training set and *x_i_* is the descriptor/bit vector for a query molecule. Typically, the threshold, *h**, is computed with Equation (10), where *m* is the number of features and *p* is the number of molecules in the training set. The feature importance was calculated using the permutation importance approach from scikit-learn with 30 repeats.

The classification models were evaluated using the following metrics: balanced accuracy, recall, precision, F1-score, Matthews correlation coefficient (MCC), and area under the ROC (receiver operating characteristics) curve (ROC_AUC). All metrics were derived from confusion matrices created from the number of true-positive (TP), true-negative (TN), false-positive (FP), and false-negative (FN) predictions. Sensitivity (true-positive rate, or recall) (Sensitivity = TP/(TP + FN)) and specificity (true-negative rate) (Specificity = TN/(TN + FP)) reflect the ability of the model to correctly classify a sample as positive (sensitivity), or negative (specificity) considering all positive data points or all negative data points, respectively. Balanced accuracy represents the average of the sensitivity and specificity, which prevents inflated performance estimates in imbalanced datasets (BA  =  (Sensitivity + Specificity)/2). Precision reflects the ability of the classifier to correctly label all positive samples as positive (Precision = TP/(TP + FP)), whereas the F1-score represents the harmonic mean between precision and recall (sensitivity), which summarizes the precision and robustness of the classifier (F1 = 2 × (Precision × Recall)/(Precision + Recall)). MCC could be seen intuitively as the summary of all categories in the confusion matrix (Appendix A). This balanced metric can be used even if the classes have different sizes. Values of MCC above 0 indicate that the classifier performed well in all four confusion-matrix categories. The ROC curve is created by plotting the fraction of true-positive rates vs. the fraction of true-negative rates at various thresholds. Additionally, the ROC_AUC value reflects the probability of the classifier to rank the randomly chosen positive example higher than the randomly chosen negative example. The values of ROC_AUC, precision, recall and F1 for multiclass models were calculated as macro-averages using a one-vs-rest approach. Additional evaluation of external performance of the classification models was performed by creating a decoy dataset with a DUD-E server, which was further concatenated to the external set [67]. For interpretation of atomic contributions and graphical interpretation of the structure–activity relationships, similarity maps were built according to the RDKit implementation of the approach proposed by Riniker et al. [51]. Chemical space projections of virtual hits from virtual screening of the SPECS database [52] were calculated using self-organizing maps, as described in previous work [20].

In vitro enzymatic evaluation of potency for the in-house library of compounds was performed using the fluorometric assay, as described elsewhere [68]. The percent of inhibition for all compounds was evaluated in triplicate at 200 µM. Details on the performed assay are provided in the Appendix A.

## 4. Conclusions

An increasing number of preclinical evidence demonstrates the potential of SIRT2 inhibitors as novel therapeutics for the treatment of a number of age-related disorders. Despite the growing interest in the development of small-molecule SIRT2 inhibitors in the past decade, none of the SIRT2 inhibitors have entered clinical trials. Currently, there is a lack of large-scale and robust structure–activity relationship models for the prediction of SIRT2 inhibitor potency and selectivity, which could greatly reduce the time and cost of developing novel inhibitors. Inspired to facilitate the discovery of novel SIRT2 inhibitors, we collected all of the currently available structure–activity information and developed a set of high-quality machine-learning models for predictions of novel inhibitor potency and selectivity. After extensive validation of the external predictive power of the models, four models were singled out: the binary RF:ECFP4 model and the regression XGBoost:ECFP4 model for the prediction of inhibitor potencies, as well as the RF:ECFP4 SIRT1/2 and DNN:descriptors SIRT2/3 models for the prediction of inhibitor selectivity. To provide the best practice in the application of the created models, a framework for the prediction of the activity/selectivity of novel compounds was defined and encoded into the Python-based application named SIRT2i_Predictor. SIRT2i_Predictor was equipped with an appealing easy-to-use web-based graphical user interface, which was aimed at enabling usage of the framework for the wider community. With automatic processing of input format (SMILES) and a demonstrated ability to rapidly and efficiently evaluate large databases of compounds on SIRT2 inhibitory potency and SIRT1–3 selectivity, SIRT2i_Predictor’s main utility is to support virtual-screening campaigns and prioritization of compounds for costly in vitro studies. Visualization functionalities, which allow for inspection of parts of molecules that contribute to the activity, make SIRT2i_Predictor a valuable resource for lead-optimization campaigns as well. Our benchmarking study indicated SIRT2i_Predictor’s complementarity to the recently published SBVS approach. A set of codes generated for database curation, model trainings, and GUI could be generalized on a number of pharmacologically relevant targets as part of the development of wider in silico platforms, which is one of our future directions.

Intending to aid future virtual-screening studies, lead-optimization studies, repurposing studies, or the integration of cheminformatics with omics data under the more complex precision-medicine pipelines, we made SIRT2i_Predictor’s GUI code with trained ML models freely available at https://github.com/echonemanja/SIRT2i_Predictor, accessed on 13 December 2022.

## Figures and Tables

**Figure 1 pharmaceuticals-16-00127-f001:**
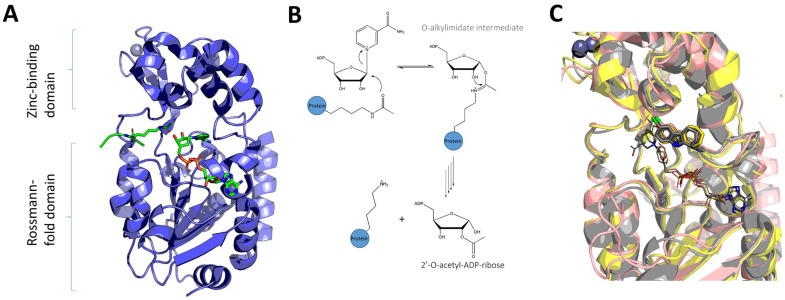
Summary of the sirtuin structures and catalytic mechanism. (**A**) Two domains of sirtuins exemplified on the structure of SIRT3 (PDB ID: 4FVT). NAD^+^ and substrate are presented in green sticks; (**B**) overview of the mechanism of sirtuin-mediated deacetylation; (**C**) problem of achieving sirtuin inhibitor selectivity exemplified through aligned structures of SIRT1 (yellow) (PDB ID: 4I5I), SIRT2 (pink) (PDB ID: 5D7P), and SIRT3 (gray) (PDB ID: 4BV3) (some parts omitted for clarity). Structurally related inhibitors (gray, pink, or yellow sticks) share the same binding mode across all isoforms. NAD^+^ and ADP–ribose are presented in gray, pink, or yellow lines.

**Figure 2 pharmaceuticals-16-00127-f002:**
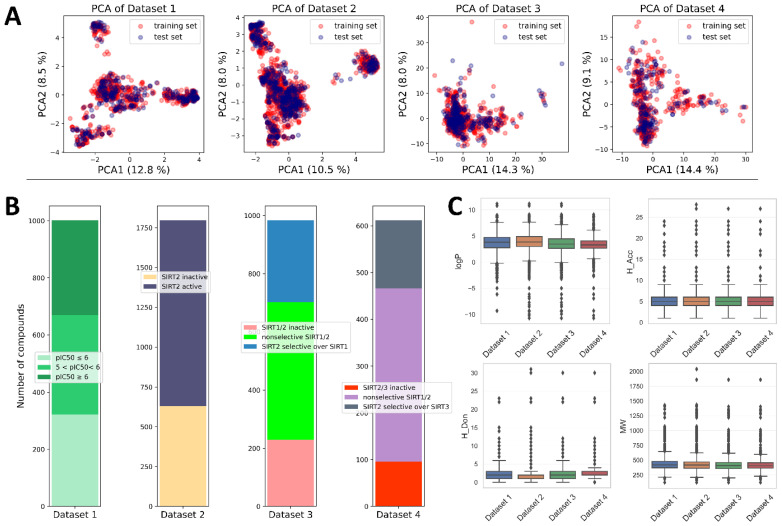
Descriptive statistics of the datasets used in the study. (**A**) PCA analysis of the chemical space of the datasets. PCA plots were calculated in accordance with the descriptors/fingerprints of the final ML models. (**B**) Distribution of data within each dataset. (**C**) Decomposition of datasets onto the Lipinski’s rule of 5.

**Figure 3 pharmaceuticals-16-00127-f003:**
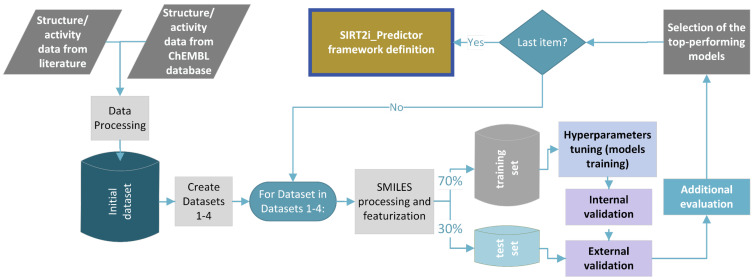
Overall design of the protocol used for generation and validation of the ML models.

**Figure 4 pharmaceuticals-16-00127-f004:**
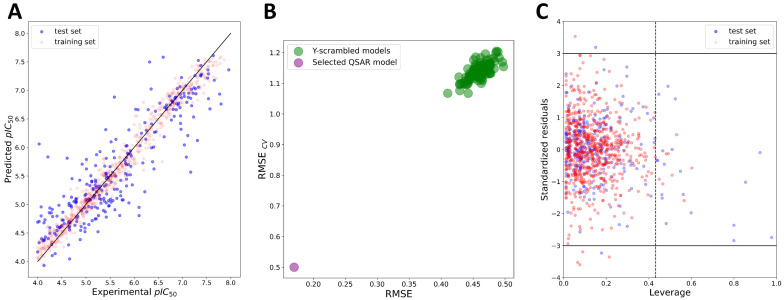
Top promising regression-based XGBoost:ECFP4 model. (**A**) Plot of experimental vs. predicted pIC50 values; (**B**) results of Y-scrambling; (**C**) applicability domain of the model. Dashed line indicates the leverage-threshold value (h*).

**Figure 5 pharmaceuticals-16-00127-f005:**
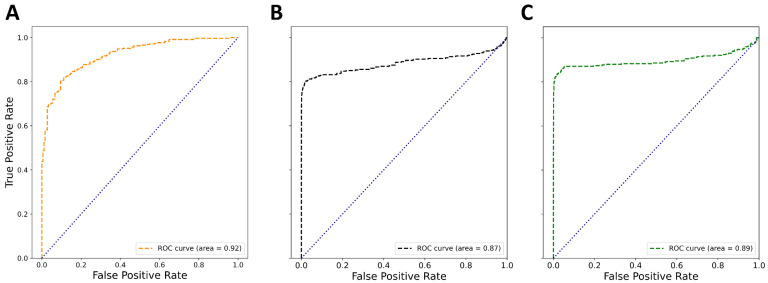
ROC curves obtained for the top-performing RF:ECFP4 binary model. (**A**) Results of external validation; (**B**) results of external validation on the decoy dataset; (**C**) results of external validation on the decoy dataset after applicability-domain corrections.

**Figure 6 pharmaceuticals-16-00127-f006:**
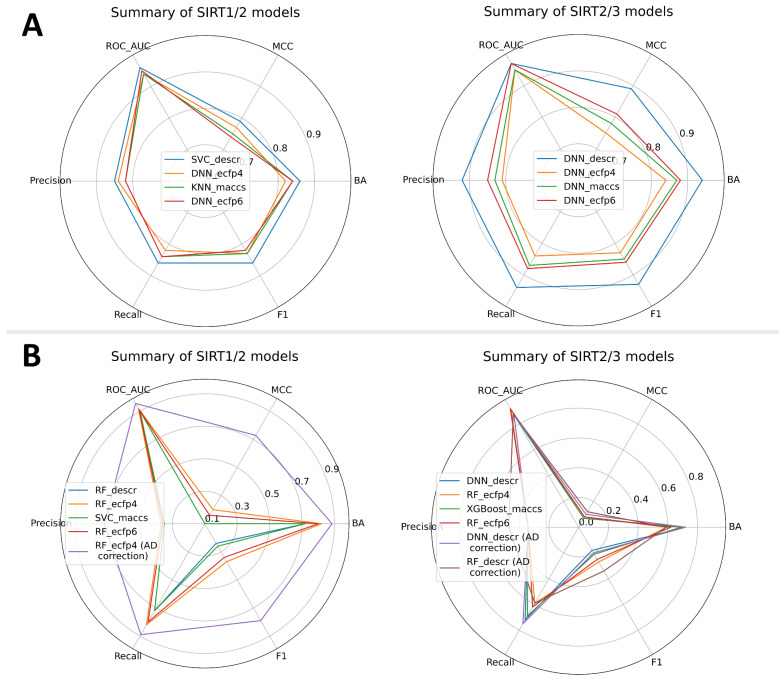
Summary of the predictive performances for multiclass selectivity models presenting the top-performing model across each feature type. (**A**) Validation parameters obtained for the external (test) set; (**B**) validation parameters obtained on the decoy set. Values of precision, recall and F1 are calculated as macro averages.

**Figure 7 pharmaceuticals-16-00127-f007:**
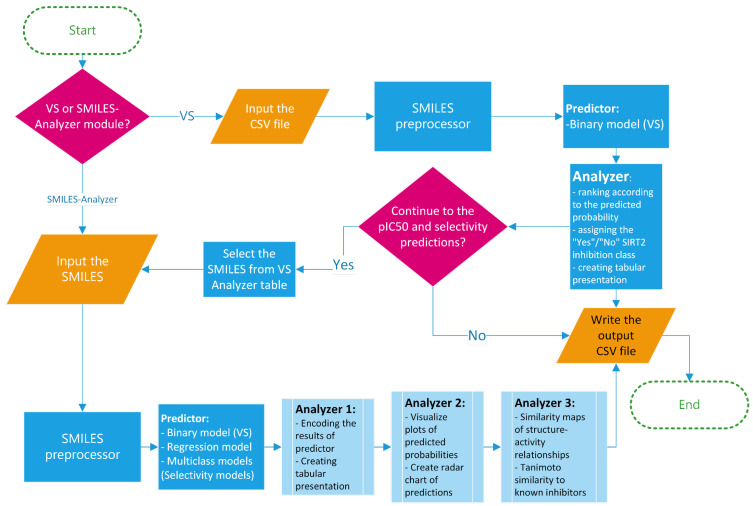
SIRT2i_Predictor framework.

**Figure 8 pharmaceuticals-16-00127-f008:**
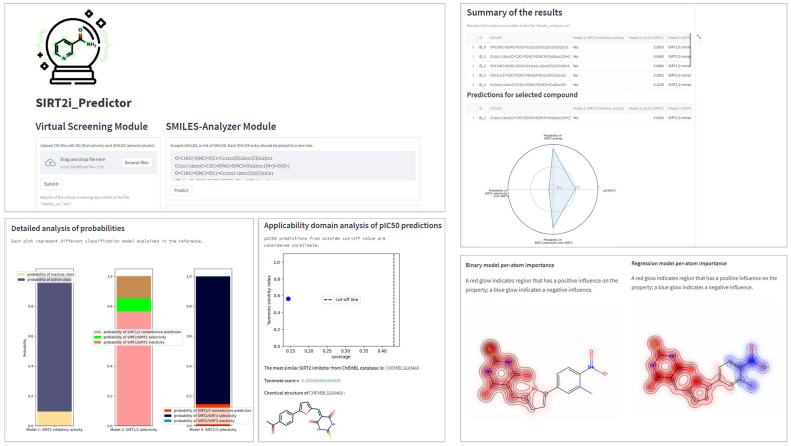
Preview of the main functionalities of SIRT2i_Predictor.

**Figure 9 pharmaceuticals-16-00127-f009:**
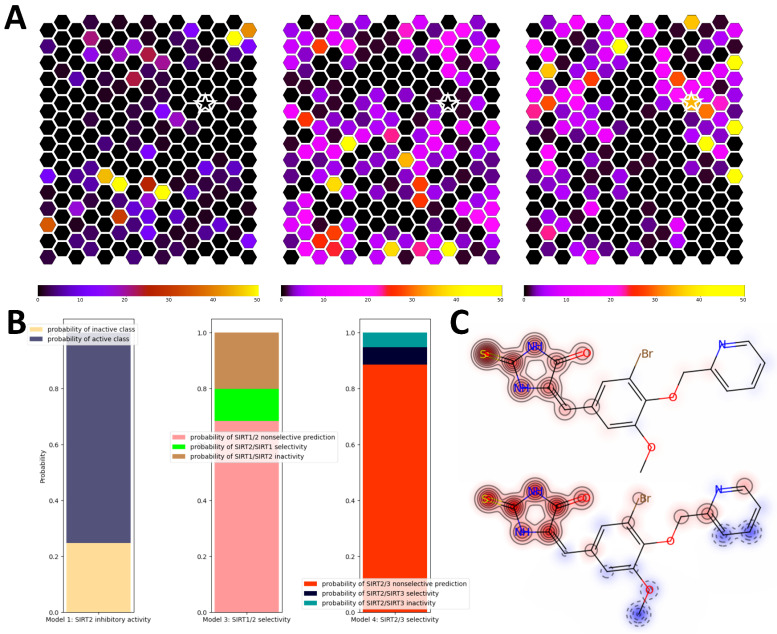
Comparison of SIRT2i_Predictor to the multi-structure SBVS approach. (**A**) Comparisons of the chemical space covered by ChEMBL SIRT2 inhibitors (left), SBVS virtual hits (middle), and SIRT2i_Predictor virtual hits (right); (**B**) analysis of probabilities obtained for virtual hits extracted from a unique portion of chemical space covered by SIRT2i_Predictor (star sign in (**A**)); (**C**) structure–activity relationships calculated using similarity maps for a binary model (upper plot) and a regression model (lower plot) for virtual hits extracted from a unique portion of chemical space (star sign in (**A**)). A red glow indicates a region that has a positive influence on the activity, whereas a blue glow indicates a negative influence.

**Table 1 pharmaceuticals-16-00127-t001:** Description of the datasets.

	Dataset 1	Dataset 2	Dataset 3	Dataset 4
No. of compounds	1002	1797	984	612
Expressed activity	pIC_50_	pIC_50_ and Inh%	Inh%	Inh%
Activity towards	SIRT2	SIRT2	SIRT1, SIRT2	SIRT2, SIRT3
Encoded activity	pIC_50_	Active, inactive	Selective, nonselective, inactive	Selective, nonselective, inactive

**Table 2 pharmaceuticals-16-00127-t002:** External validation parameters of regression QSAR models.

ML Algorithm	Molecular Feature	$R_{ext}^{2}$	*RMSE_ext_*	${\bar{r}}_{m}^{2}\;$	$\Delta r_{m}^{2}$	$Q_{F1}^{2}$	$Q_{F2}^{2}\;$	$Q_{F3}^{2}$	*CCC*
RF	Descriptors	0.7	0.55	0.52	0.27	0.7	0.7	0.7	0.81
	ECFP4	0.75	0.5	0.6	0.23	0.75	0.75	0.75	0.85 ^a^
	MACCS	0.71	0.53	0.55	0.26	0.71	0.71	0.71	0.82
	ECFP6	0.77	0.48	0.62	0.21	0.77	0.77	0.76	0.86 ^a^
SVR	Descriptors	0.62	0.61	0.44	0.31	0.62	0.62	0.62	0.77
	ECFP4	0.74	0.51	0.63	0.13 *	0.74	0.74	0.73	0.84
	MACCS	0.68	0.57	0.55	0.21	0.68	0.68	0.68	0.81
	ECFP6	0.74	0.51	0.63	0.18 *	0.74	0.74	0.74	0.86 ^a^
XGBoost	Descriptors	0.67	0.58	0.53	0.25	0.68	0.68	0.68	0.82
	ECFP4	0.75 (0.79) ^b^	0.5 (0.46) ^b^	0.64 (0.7) ^b^	0.17 (0.17) *^,b^	0.74 (0.75) ^b^	0.74 (0.75) ^b^	0.74 (0.74) ^b^	0.86 (0.86) ^a,b^
	MACCS	0.71	0.53	0.58	0.24	0.7	0.7	0.7	0.82
	ECFP6	0.73	0.52	0.62	0.2	0.73	0.73	0.73	0.87 ^a^
KNN	Descriptors	0.68	0.56	0.56	0.23	0.68	0.68	0.68	0.86 ^a^
	ECFP4	0.74	0.51	0.64	0.13 *	0.74	0.74	0.74	0.87 ^a^
	MACCS	0.6	0.63	0.47	0.16 *	0.6	0.6	0.6	0.79
	ECFP6	0.76 (0.77) ^b^	0.49 (0.48) ^b^	0.66 (0.68) ^b^	0.12 (0.11) *^,b^	0.76 (0.76) ^b^	0.76 (0.76) ^b^	0.76 (0.76) ^b^	0.87 (0.87) ^a,b^
DNN	Descriptors	0.66	0.58	0.57	0.03 *	0.66	0.66	0.66	0.81
	ECFP4	0.74	0.51	0.63	0.18 *	0.73	0.73	0.73	0.84
	MACCS	0.68	0.56	0.56	0.16 *	0.68	0.68	0.67	0.80
	ECFP6	0.73	0.52	0.63	0.17 *	0.73	0.73	0.73	0.81
Criteria		>0.6		>0.5	<0.2	>0.7	>0.7	>0.7	>0.85

* Marks the values within the threshold $\Delta r_{m}^{2}$ < 0.2. ^a^ Marks the values above the proposed threshold *CCC* > 0.85. ^b^ Number in brackets represent values after applicability domain corrections.

**Table 3 pharmaceuticals-16-00127-t003:** External validation parameters of the binary classification models.

ML Algorithm	Molecular Feature	BA	MCC	ROC_AUC	Precision ^a^	Recall ^a^	F1 ^a^
RF	Descriptors	0.88	0.74	0.94	0.86	0.88	0.87
	ECFP4	0.84	0.66	0.92	0.82	0.84	0.83
	MACCS	0.82	0.62	0.91	0.8	0.82	0.81
	ECFP6	0.85	0.68	0.92	0.83	0.85	0.84
SVR	Descriptors	0.88	0.74	0.95	0.87	0.88	0.87
	ECFP4	0.81	0.63	0.9	0.82	0.81	0.82
	MACCS	0.8	0.59	0.87	0.79	0.8	0.79
	ECFP6	0.79	0.62	0.9	0.83	0.79	0.81
XGBoost	Descriptors	0.86	0.72	0.94	0.85	0.86	0.85
	ECFP4	0.81	0.62	0.91	0.8	0.81	0.81
	MACCS	0.8	0.6	0.9	0.8	0.8	0.8
	ECFP6	0.81	0.62	0.91	0.81	0.81	0.81
KNN	Descriptors	0.79	0.56	0.88	0.77	0.79	0.77
	ECFP4	0.82	0.62	0.9	0.8	0.82	0.81
	MACCS	0.82	0.62	0.88	0.8	0.82	0.81
	ECFP6	0.84	0.65	0.91	0.81	0.84	0.82
DNN	Descriptors	0.89	0.75	0.94	0.85	0.86	0.86
	ECFP4	0.83	0.65	0.91	0.8	0.81	0.8
	MACCS	0.8	0.58	0.89	0.8	0.8	0.8
	ECFP6	0.82	0.64	0.9	0.79	0.82	0.8

^a^ Average values across classes are presented.

**Table 4 pharmaceuticals-16-00127-t004:** The predictive performance parameters of the binary models on the decoy dataset.

ML Algorithm	Molecular Feature	BA	MCC	ROC_AUC	Precision ^a^	Recall ^a^	F1 ^a^	EF_05%_	EF_1%_	EF_2%_	EF_5%_
RF	Descriptors	0.68	0.09	0.87	0.51	0.68	0.35	0.63	0.67	0.68	0.73
	ECFP4	0.81 (0.9) ^b^	0.19 (0.52) ^b^	0.87 (0.89) ^b^	0.53 (0.67) ^b^	0.81 (0.9) ^b^	0.49 (0.73) ^b^	0.74 (0.74) ^b^	0.74 (0.74) ^b^	0.76 (0.76) ^b^	0.77 (0.8) ^b^
	MACCS	0.66	0.08	0.82	0.51	0.66	0.35	0.55	0.56	0.59	0.62
	ECFP6	0.75	0.14	0.87	0.52	0.75	0.43	0.72	0.74	0.76	0.78
SVR	Descriptors	0.69	0.1	0.89	0.51	0.69	0.36	0.43	0.56	0.62	0.71
	ECFP4	0.46	−0.06	0.8	0.48	0.46	0.05	0.75	0.75	0.75	0.76
	MACCS	0.62	0.06	0.83	0.51	0.62	0.32	0.39	0.61	0.68	0.74
	ECFP6	0.47	−0.07	0.8	0.47	0.47	0.03	0.76	0.76	0.77	0.77
XGBoost	Descriptors	0.71	0.11	0.85	0.51	0.71	0.39	0.41	0.44	0.48	0.54
	ECFP4	0.74	0.13	0.87	0.52	0.74	0.42	0.35	0.39	0.43	0.52
	MACCS	0.64	0.07	0.73	0.51	0.64	0.32	0	0	0.02	0.2
	ECFP6	0.71	0.11	0.85	0.51	0.71	0.39	0.37	0.38	0.44	0.5
KNN	Descriptors	0.66	0.08	0.76	0.51	0.66	0.37	0.09	0.23	0.26	0.29
	ECFP4	0.72	0.12	0.8	0.52	0.72	0.41	0	0	0	0
	MACCS	0.64	0.07	0.75	0.51	0.64	0.33	0	0	0	0
	ECFP6	0.72	0.11	0.8	0.52	0.72	0.41	0	0	0	0
DNN	Descriptors	0.72	0.12	0.8	0.51	0.71	0.38	0	0	0	0
	ECFP4	0.73	0.13	0.84	0.52	0.73	0.43	0.1	0.25	0.32	0.41
	MACCS	0.69	0.1	0.79	0.51	0.62	0.29	0.04	0.08	0.17	0.23
	ECFP6	0.67	0.09	0.81	0.51	0.67	0.38	0.17	0.25	0.34	0.43

^a^ Average values across classes are presented. ^b^ Number in brackets represents value after applicability domain corrections.

## Data Availability

Data are contained within the article and Appendix A. The code for the developed software, together with the machine-learning models generated through the study, are freely available at https://github.com/echonemanja/SIRT2i_Predictor, accessed on 13 December 2022.

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
