# Peer review of "SIRT2i_Predictor: A Machine Learning-Based Tool to Facilitate the Discovery of Novel SIRT2 Inhibitors"

_pharmaceuticals, 2023, doi:10.3390/ph16010127_

Round 1

Reviewer 1 Report

The computational study of inhibitors as novel therapeutics for treatment of number of age-related disorders is of great importance for the professional community. Specifically, presented in this study large-scale and robust structure-activity relationship models for prediction of SIRT2 inhibitors potency and selectivity which will greatly reduce time and costs of development of novel inhibitors. This work provides the best practice in application of created models, a framework for prediction of activity/selectivity of novel compounds. The model was defined and encoded into the Python based application named SIRT2i_Predictor. SIRT2i_Predictor was equipped with the appealing easy-to-use web-based graphical user interface. This web-based interface will significantly increase usability of the framework to the wider community.

Reviewer 2 Report

Comments and queries

1.     It is recommended to rewrite the abstract giving more emphasis on machine learning techniques and their relevance to the current topic and need of the hour.

2.     Authors need to mention clearly about the research gaps in the search for sirtuin inhibitors or inhibitors of selective isoforms.

3.     Authors are advised to introduce structural features of the target of interest, a diagram or two will suffice. Also, about binding mechanism and type of inhibition mechanism.

4.     Are all the compounds under study have been carried out by the same assay protocol? ( Ex: fluorescence, colorimetry or ELISA etc.). This is required for the validity of models generated.

5.     Reasons for classifying data into 4 sets?

6.     Please list the important molecular descriptors used for screening.

7.     Explain the rational behind the usage of MACCS and ECFP fingerprints. Any corelation with the structures under consideration?

8.     Have you used any AI (Artificial Intelligence ) based applications in any one of the screening methods. ?

General Comments to Authors

1.     The present work is commendable as you have integrated various cheminformatics approaches.

2.     Large number of data sets are screened using variety of descriptors and features, this makes the model more reliable.

3.     Statistics are well presented and clearly defined
